# Autoignition of Methane–Hydrogen Mixtures below 1000 K

Vladimir Arutyunov [1,2,*], Andrey Belyaev [1], Artem Arutyunov [1,3], Kirill Troshin [1] and Aleksey Nikitin [1,2]

1   N.N. Semenov Federal Research Center for Chemical Physics, Russian Academy of Sciences, Kosygina 4, Moscow 119991, Russia
2   Institute of Problems of Chemical Physics, Russian Academy of Sciences, Semenova 1, Moscow 142432, Russia
3   Faculty of Computational Mathematics and Cybernetics, Shenzhen MSU-BIT University, Shenzhen 518172, China
*   Correspondence: v_arutyunov@mail.ru

**Abstract:** In the range of 800–1200 K, both experiments and kinetic modeling demonstrate a significant difference in the dependence of the ignition delay time of methane and hydrogen on pressure and temperature, with the complex influence of these parameters on the autoignition delay time of methane–hydrogen–air mixtures. In connection with the prospects for the widespread use of methane–hydrogen mixtures in energy production and transport, a detailed analysis of their ignition at temperatures below 1000 K, the most important region from the point of view of their practical application, is carried out. It is shown that such a complex behavior is associated with the transition in this temperature range from low-temperature mechanisms of oxidation of both methane and hydrogen, in which peroxide radicals and molecules play a decisive role, to high-temperature mechanisms of their oxidation, in which simpler radicals dominate. A kinetic interpretation of the processes occurring in this case is proposed.

**Keywords:** methane; hydrogen; methane–hydrogen mixtures; autoignition delay time; kinetic modeling

## 1. Introduction

Serious concerns from the world community, politicians and industry circles about the negative impact of energy and transport on the global climate, and active efforts to reduce the growth of carbon dioxide concentration in the atmosphere [1] have stimulated interest in low-carbon energy sources and energy carriers. A transition to renewable energy sources is considered as the main direction, but in the near future their contribution to global energy is unlikely to be sufficient to significantly affect the emission of carbon dioxide into the atmosphere [2]. Therefore, more and more attention is being paid to low-carbon energy carriers, for example, ammonia [3]. However, there is little doubt that hydrogen and hydrogen-containing mixtures continue to be widely used as low-carbon energy carriers.

The use of methane–hydrogen mixtures with different hydrogen content at the initial stage of the transition to "hydrogen energy" will make it possible to circumvent many complex problems concerning the production, storing, transportation and distribution of hydrogen [4–7]. Along with hydrogen, such mixtures can be used to power traditional internal combustion engines (ICE) [8–11], the energy efficiency of using hydrogen in which is not much inferior to the efficiency of its use in fuel cells [9]. In addition, while ICE is a cheaper and more reliable source of energy than fuel cells, the use of hydrogen and mixtures containing it in ICE is technically more developed. Additionally, some engine tests have demonstrated that using hydrogen-enriched natural gas widens the lean burn operation range, while reducing unburned hydrocarbon and carbon dioxide emissions [12]. Due to progress in the development of ICE, it is expected that by 2045 their fuel efficiency will nearly reach that of fuel cells [13]. Therefore, the use of hydrogen and methane–hydrogen mixtures to power the ICE can become a natural transition link between modern liquid or gas-fueled ICE and future fuel cell-based transport [14].

However, practical applications of methane–hydrogen mixtures require a detailed study of the possibility and conditions of their use for operation with existing power equipment. To optimize the composition of gas mixtures and the operating modes of existing power equipment on such mixtures, as well as to ensure the safety of their storage, transportation and utilization, it is necessary to study the conditions of their ignition and the parameters characterizing the process of their combustion.

The basic laws of the combustion of hydrogen and methane were established a long time ago [15]; however, there are still many white spots and a large field for research that needs to be carried out to ensure the possibility of the widespread use of methane–hydrogen mixtures. Among the most important parameters determining the optimal conditions and safety of the utilization of methane–hydrogen mixtures are the autoignition delay time and laminar burning velocity. A large body of work has been devoted to the study of these parameters for hydrogen and methane, but there are only a few experimental investigations on the ignition characteristics and chemistry for methane–hydrogen mixtures. Note also that almost all studies of the ignition delay time were carried out in shock tubes or in rapid compression machines [16–25] at temperatures above 1000 K. However, for the safe utilization of such mixtures, it is desirable to obtain data on their ignition at temperatures as low as possible. In addition, since the autoignition of the working mixture in the internal combustion engine occurs at relatively low temperatures, ~500–900 K [26], the optimization of the performance of methane–hydrogen mixtures in ICEs requires the determination of the ignition delay time within this temperature range.

## 2. Effect of Hydrogen on the Autoignition of Methane in the Transition Temperature Region

Studying the autoignition of methane–hydrogen mixtures at temperatures below 1000 K is complicated by the fact that, in this temperature range, the mechanisms of the oxidation of methane and hydrogen change dramatically. Within a narrow temperature range 900–1000 K, the low-temperature mechanisms of their oxidation, in which peroxide compounds and radicals play a significant role, changes to the high-temperature oxidation mechanisms, in which reactions involving $H^\bullet$, $O^{\bullet\bullet}$, $OH^\bullet$, and $CH_3^\bullet$ dominate. These changes in the mechanisms strongly affect the autoignition of hydrogen and methane, which usually begins by a low-temperature mechanism followed by transition to high-temperature oxidation. For methane oxidation in this temperature range, this gives rise to various nonlinear effects, such as negative temperature coefficient (NTC) of the oxidation rate, cool flames, the inhibition of methane oxidation by oxygen, and a number of others [27]. During the oxidation of methane–hydrogen mixtures, these changes in the oxidation mechanisms of both compounds overlap, leading, depending on the ratio of methane and hydrogen in the mixture, pressure, initial temperature and other conditions to a complex pattern of the observed behavior, the study and interpretation of which is devoted to in this work.

The complex influence of hydrogen on the autoignition of methane has been known for a long time. Gersen et al. [20] measured the ignition delay of methane–hydrogen mixtures in a rapid compression machine under stoichiometric conditions at pressures from 1.5 to 7.0 MPa, temperatures from 950 to 1060 K, and hydrogen mole fractions from 0% to 100%. Their results showed that the promotion effect of hydrogen is only marginal for hydrogen fraction below 20%, while the ignition delay decreased remarkably when the hydrogen fraction is over 50%. Furthermore, the promotion of ignition is boosted by increasing temperature but is suppressed by increasing pressure.

It was found [23] that at $T > 1000$ K, the measured ignition delay time agrees well with theoretical predictions, while at $T < 1000$ K, this parameter turns out to be substantially smaller than the calculated value, with the difference reaching three orders of magnitude at ~800 K.

Experiments behind reflected shock waves at temperatures from 1000 to 2000 K and pressures from 0.5 to 2.0 MPa in conjunction with kinetic simulations [24] have shown

that the pressure dependence of the ignition delay time for methane–hydrogen mixtures at hydrogen fractions less than 40% resembles that of methane: the ignition delay decreases with increasing pressure. At a hydrogen fraction of 60%, the promotion effect of pressure on the ignition of methane–hydrogen mixtures was negligibly small. At hydrogen fractions equal or greater than 80%, the ignition response resembled that of hydrogen in that the ignition delay exhibited a complex dependence on pressure and a two-step transition in the global activation energy. Kinetic simulations using a NUI Galway mechanism demonstrated excellent agreement with these results.

According to the hydrogen fraction, the authors of [25] identified three ignition regimes: at $[H_2] \leq 40\%$, methane chemistry dominates; at $[H_2] = 60\%$, combined chemistry of methane and hydrogen manifests itself; and $[H_2] \geq 80\%$, hydrogen chemistry is leading. Note also that a significant difference was observed in the temperature dependences of the autoignition delay time for methane and hydrogen at high and low temperatures. While at temperatures above 1250 K, the regular Arrhenius dependence is observed in both cases, at lower temperatures this dependence for both methane and hydrogen exhibit a complex behavior, with a significant change in the activation energy of the ignition delay time (Figure 1).

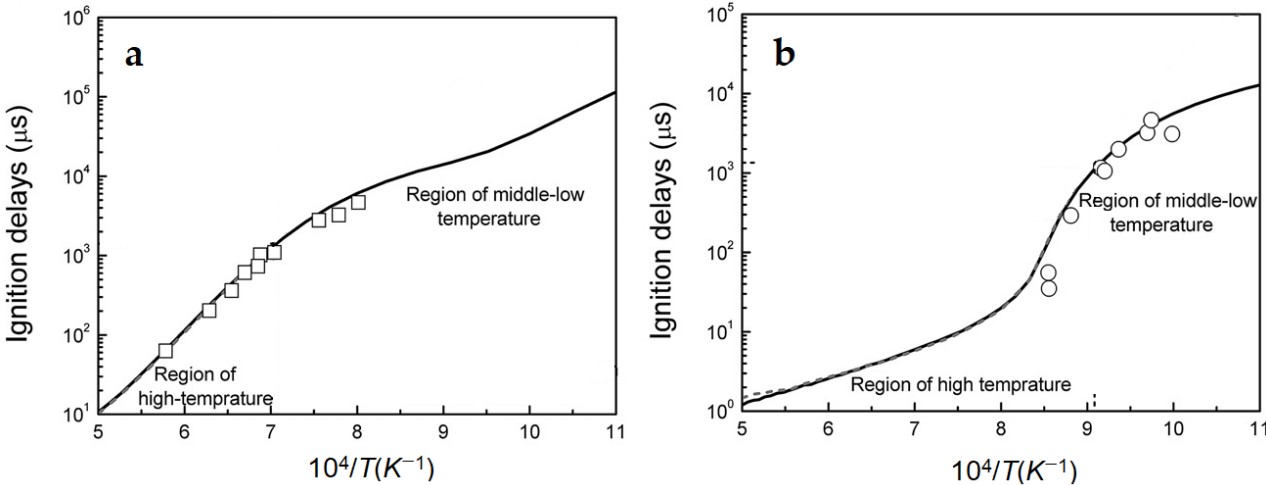

**Figure 1.** Temperature dependence of ignition delays for methane (**a**) and hydrogen (**b**). Symbols—experimental data, lines—kinetic calculation results (adapted from [25]).

Due to the importance of information on the autoignition and combustion characteristics of methane–hydrogen mixtures in the temperature range of 800–1000 K for their safe handling and their utilization in ICE, in this study, the delays of autoignition of methane–hydrogen mixtures in this range were experimentally determined and a kinetic analysis of the process of their autoignition was carried out.

### 3. Experimental Investigation of the Autoignition Delay Time of Methane–Hydrogen Mixtures

In this study, an experimental investigation of the autoignition delay time of methane–hydrogen mixtures was carried out in a static-type setup with a closed reaction vessel by the high-pressure bomb method described in detail in [28,29]. The reactor was a thick-walled heated cylindrical vessel made of stainless-steel, with a diameter and height of 10 cm. Mixtures of methane–hydrogen–air of a given composition were prepared in high-pressure steel cylinders according to the partial pressures of the components. The prepared mixtures were allowed to stay in these cylinders for at least 48 h to provide complete mixing of the gas components. The evacuated reactor was heated to a desired temperature $T_0$ and filled with a test mixture to a desired initial pressure $P_0$ through an electromagnetic valve synchronized with the registration system. The pressure in the reactor was recorded by a sensor with a normal frequency of 5–8 kHz. The autoignition delay time was defined as the

time from the moment of pressure equalization after mixture admission into the reactor to the moment of a sharp pressure rise as a result of its autoignition. Typical pressure change curves are presented in [28,29]. The autoignition delay time that can be measured by this method is limited from below by the time of mixture inlet in the reactor and equalizing of its temperature, which is ~0.2 s, and from above by a period of about 20 s, exceeding which can lead to an uncontrolled change in the composition of the mixture and the state of the reactor surface. Due to the stochastic nature of the autoignition, uncontrolled changes in the state of the inner surface of the reactor and the complex gas dynamics of the mixture injection process, the variation in autoignition delay time between successive experiments can reach 30%. However, a sufficient number of experiments and the strong temperature dependence of the autoignition delay time allow us to obtain a fairly reliable array of data to determine the activation energy.

The autoignition delay times of stoichiometric methane–hydrogen–air mixtures in the temperature range of 850–1000 K at initial pressures of $P_0 = 1$ and 3 atm and hydrogen concentrations in the mixture from 0 to 50% were investigated. At higher hydrogen concentrations, the autoignition delay time for this temperature range is shorter than the low boundary of reliable measurements. The obtained temperature dependence of the autoignition delay time for stoichiometric methane–hydrogen–air mixtures (Figure 2) is well described by the Arrhenius expression

$$\tau = A \exp{(E_a/RT)},$$

where $E_a$ is the effective activation energy, and $A$ is the pre-exponential factor.

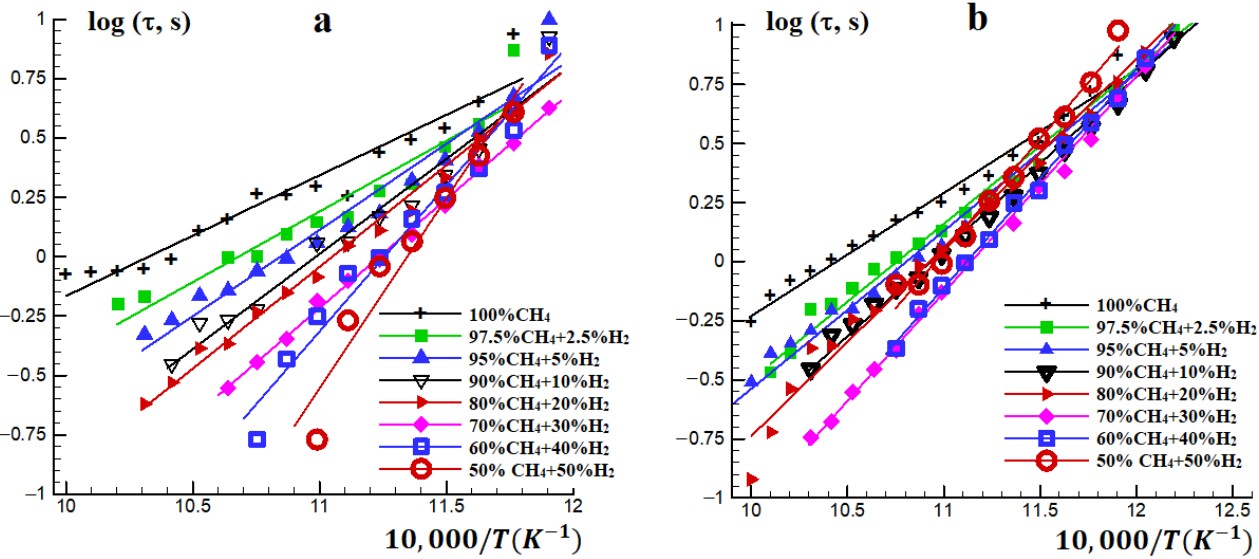

**Figure 2.** Temperature dependence of the autoignition delay time for stoichiometric methane–hydrogen-air mixtures at $P_0 = $ (**a**) 1 and (**b**) 3 atm.

Note that at high temperatures, the autoignition delay time shortens significantly with increasing hydrogen concentration, i.e., hydrogen promotes the ignition of methane. However, at lower temperatures $T \approx 850$ K, the promotion effect is weak, if any (Figure 2). An increase in pressure narrows the range of changes in the effective activation energy of the autoignition delay time with a change in the hydrogen concentration in the mixture.

The effective activation energy of the autoignition delay time for methane–hydrogen mixtures increases with the hydrogen concentration, being accompanied by a decrease in the pre-exponential factor $A$. At ~900 K and at $P_0 = 1$ atm, the increase in the hydrogen concentration in the mixture from 0 to 50% results in a threefold increase in the effective activation energy: from 23.4 to 73.7 kcal/mol (Figure 3). Such a significant change in the effective activation energy of the autoignition delay time is indicative of a serious

change in the mechanism of autoignition in this temperature range. This effect of hydrogen addition on the autoignition of methane distinguishes it sharply from that addition of $C_2$–$C_6$ alkanes on it, for which, regardless of the added alkane concentration, the effective activation energy for the autoignition delay time remain practically unchanged, within $40 \pm 10$ kcal/mol [28–30].

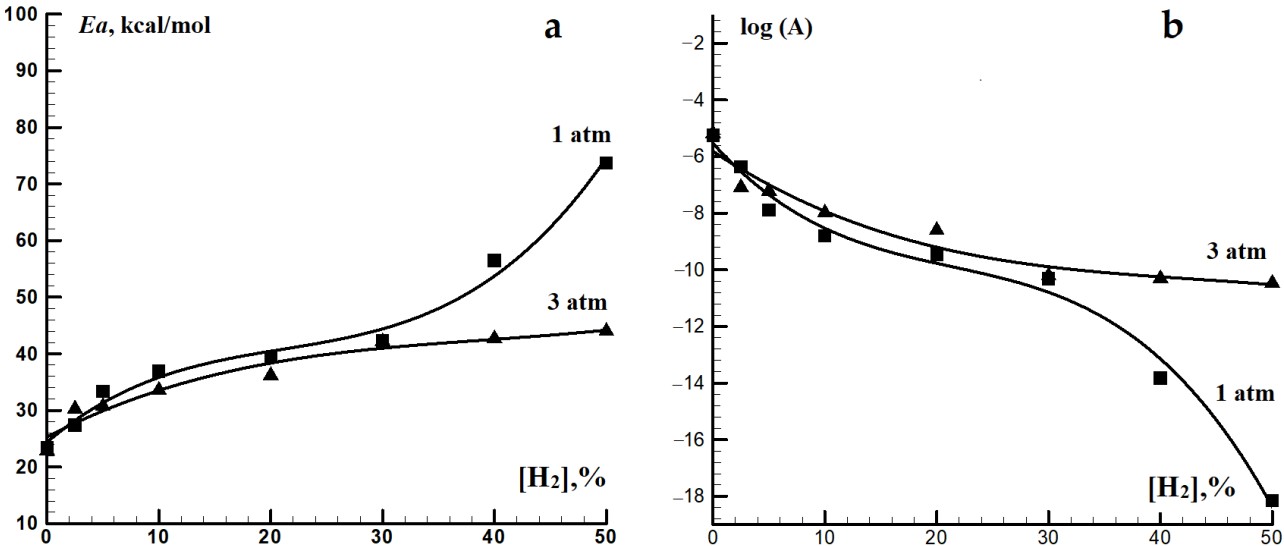

**Figure 3.** Dependence of the effective activation energy *E*a (**a**) and the pre-exponential factor *A* (**b**) in the Arrhenius expression for the autoignition delay time for methane–hydrogen-air mixtures on the concentration of hydrogen at $P_0 = 1$ and 3 atm.

A consequence of the autoignition delay time activation energy for methane–hydrogen mixtures increasing with the hydrogen concentration is an enhancement in their sensitivity to temperature changes and, consequently, a decrease in their detonation resistance, since not only is the value of the ignition delay itself important, but also the sensitivity of the fuel to changes in temperature, concentration, and pressure. Figure 3 shows that at hydrogen concentrations in the mixture below 30%, pressure has little effect on the change in the activation energy for the autoignition delay time. At higher hydrogen concentrations, an increase in pressure makes the effect of hydrogen on the activation energy of the autoignition delay time of methane–hydrogen mixtures less pronounced.

## 4. Kinetic Modeling of Autoignition Delay of Methane-Hydrogen Mixtures

Experimental results obtained even in a limited range of hydrogen concentrations ($\leq 50\%$) indicate the complex influence of hydrogen concentration on the autoignition of methane–hydrogen mixtures at temperatures below 1000 K. To obtain more detailed information on this influence, simulations of the autoignition delay times of stoichiometric $CH_4$–$H_2$–air mixtures with different hydrogen concentrations in them were carried out. The kinetic mechanism NUI Galway [31,32] was used in calculations, which proved to be the most adequate for describing these processes [30].

Since the mechanism of hydrogen oxidation is a model object of research and has been studied in sufficient detail, ignition delays for hydrogen–air mixtures calculated by the mechanism [31,32] were compared with calculations based on more specialized mechanisms of hydrogen oxidation [33–35]. Analysis shows that for the conditions we consider, the mechanism from [31,32] adequately describes the process of autoignition of hydrogen–air mixtures, without revealing any noticeable discrepancies with the mechanisms [33–35]. The obtained values of the autoignition delay time for temperatures 800–1000 K are presented in Table 1, while the corresponding temperature dependences are displayed in Figure 4.

**Table 1.** Calculated autoignition delay time lg$\tau$ (s) of stoichiometric $CH_4$-$H_2$-air mixtures.

| [$H_2$], % | Temperature, K | | | | |
|---|---|---|---|---|---|
| | 1000 | 950 | 900 | 850 | 800 |
| 0 | 0.090 | 0.459 | 0.852 | 1.276 | 1.755 |
| 10 | −0.348 | 0.009 | 0.401 | 0.866 | 1.440 |
| 20 | −0.710 | −0.338 | 0.100 | 0.648 | 1.319 |
| 30 | −1.049 | −0.654 | −0.155 | 0.490 | 1.240 |
| 40 | −1.378 | −0.959 | −0.377 | 0.371 | 1.180 |
| 50 | −1.719 | −1.262 | −0.576 | 0.277 | 1.127 |
| 60 | −2.084 | −1.600 | −0.767 | 0.197 | 1.076 |
| 70 | −2.506 | −2.013 | −0.971 | 0.123 | 1.025 |
| 80 | −2.971 | −2.479 | −1.215 | 0.048 | 0.967 |
| 90 | −3.411 | −2.936 | −1.425 | −0.032 | 0.901 |
| 100 | −3.690 | −3.309 | −1.554 | −0.122 | 0.820 |

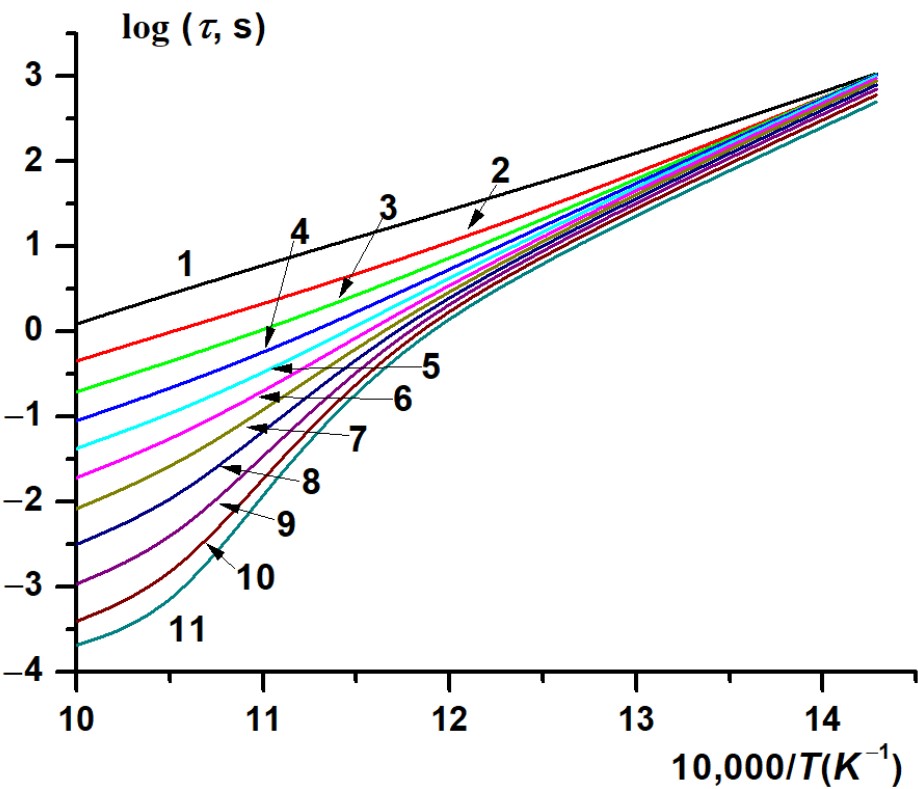

**Figure 4.** Calculated temperature dependence of the autoignition delay time of stoichiometric $CH_4$-$H_2$-air mixtures on the initial temperature at different concentrations [$H_2$] (%): 0 (*1*), 10 (*2*), 20 (*3*), 30 (*4*), 40 (*5*), 50 (*6*), 60 (*7*), 70 (*8*), 80 (*9*), 90 (*10*), 100 (*11*). $P_0$ = 1 atm.

The calculated dependences demonstrate a complex influence of hydrogen on the autoignition delay time for methane. At hydrogen concentrations in the mixture up to 40%, the temperature dependence of the autoignition delay time is closely described by the Arrhenius expression (Figure 4, curves *1–5*). However, at higher hydrogen concentrations, the dependence ceases to be of Arrhenius type (Figure 4, curves *6–11*), with the temperature dependence of the effective activation energy for the ignition delay time passing through a maximum at ~900 K (Figure 5, curves *3–5*). While the effective activation energy of the autoignition delay of methane itself (Figure 5, curve *1*) remains almost constant,

~30 kcal/mol, over the entire temperature range covered, the effective activation energy of the autoignition delay time of hydrogen and mixtures with its high content increases near $T \approx 900$ K by about a factor of 3–4, compared to the effective activation energy of the autoignition delay time at lower and higher temperatures (Figure 5, curves *4*, *5*).

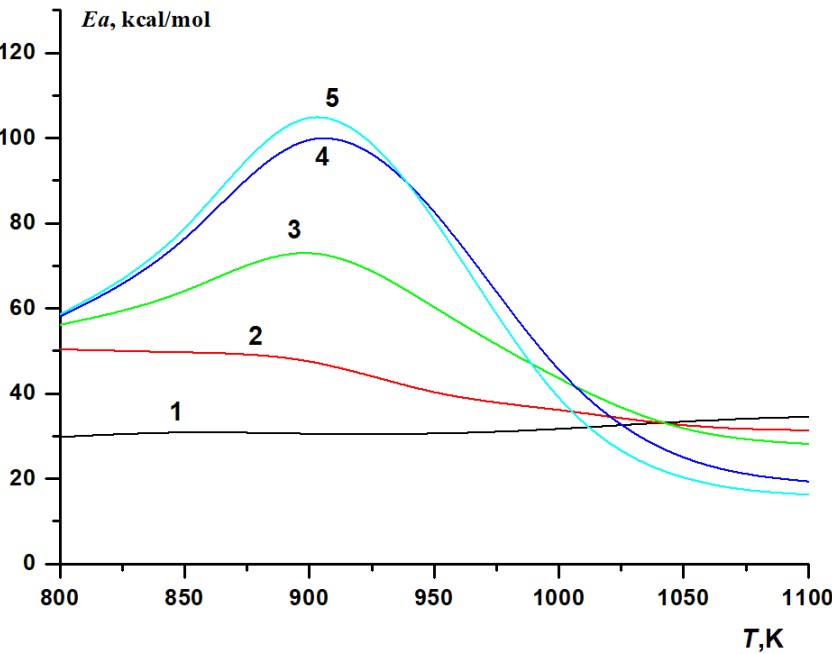

**Figure 5.** Calculated temperature dependence of the effective activation energy *Ea* for the autoignition delay time of stoichiometric $CH_4$–$H_2$–air mixtures on the initial temperature at various hydrogen concentrations (%): 0 (*1*), 40 (*2*), 70 (*3*), 90 (*4*), 100 (*5*). $P_0 = 1$ atm.

It should also be noted that in low temperatures, $T < 850$ K, the effective activation energy for the autoignition delay time of hydrogen and hydrogen-containing mixtures is higher than the effective activation energy of the autoignition delay time for methane (Figure 5). Possible reasons for this will be discussed below. For mixtures with a hydrogen concentration of up to 40 vol%, the effective activation energy of the autoignition delay time decreases monotonically with increasing temperature, becoming lower than that for methane at $T > 1100$ K (Figure 5, curve *2*). For methane–hydrogen mixtures with a high hydrogen content, as well as for hydrogen itself, the effective activation energy of the autoignition delay time passes through a maximum near $T = 900$ K, but at $T > 1100$ K it also becomes lower than that for methane. At the same time, the higher the hydrogen content in the mixture, the lower it is (Figure 5).

## 5. Influence of Hydrogen Concentration on the Laminar Burning Velocity of Methane–Hydrogen Mixtures

The addition of hydrogen to methane–air mixtures increases their laminar burning velocity and expands the flammability limits [36–49], although these additives must be significant for a noticeable effect. For example, it was shown [47] that the addition of 10 or 20% hydrogen to methane has a weak effect on the laminar velocity of its flame, but significantly expands the lean flame propagation limit. An increase in the initial pressure leads to a decrease in the burning velocity of both methane–air mixtures and methane–hydrogen–air mixtures, with the decrease for hydrogen-free mixtures being more noticeable. The lean flame propagation limit expands with increasing initial pressure.

To obtain a more detailed picture of the effect of hydrogen concentration on the ignition and combustion of methane–hydrogen mixtures, we calculated their laminar burning velocity at various initial temperatures. The effect of the initial temperature on the laminar burning velocity of methane–hydrogen–air mixtures was investigated.

Calculations were carried out using the mechanism [31,32]; however, for comparison, in some cases, calculations were also carried out using the global kinetic mechanism of methane combustion, which includes only 10 elementary reactions involving 9 components ($CH_4$, $O_2$, CO, $CO_2$, $H_2$, $H_2O$, $N_2$, NO and the formally introduced radical $HO_{0.5}$) [50,51]. This mechanism has been repeatedly tested when calculating the processes of autoignition and propagation of laminar flames. Under these conditions, it describes the process of combustion quite satisfactorily at low hydrogen concentrations, but at high concentrations it becomes unusable, since it does not reflect changes in the mechanism of hydrogen combustion in this temperature range. Figure 6a compares the published experimental data on the dependence of the laminar burning velocity in stoichiometric methane–hydrogen–air mixtures on the hydrogen concentration and our calculation results.

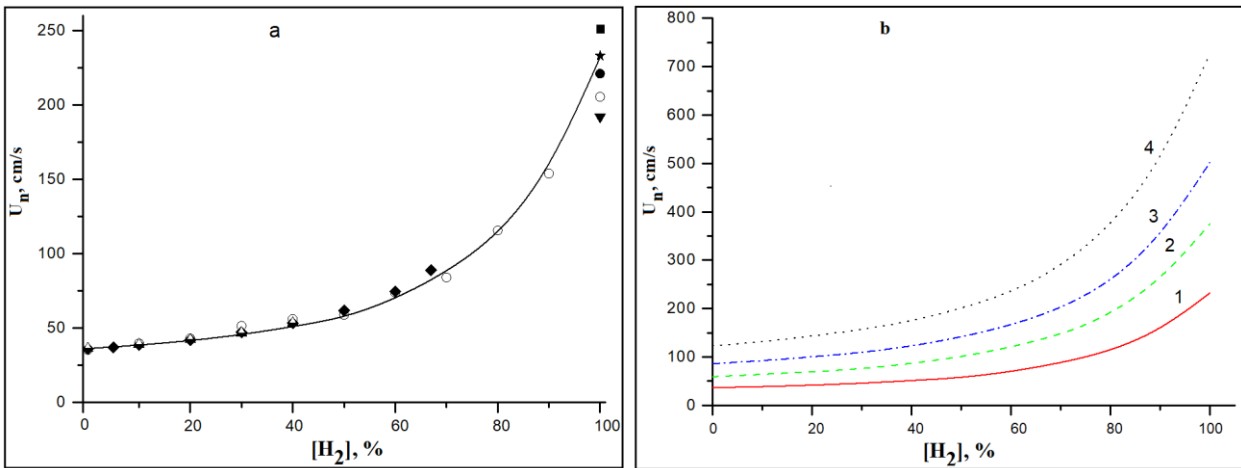

**Figure 6.** Dependence of laminar burning velocity $Un$ in stoichiometric of $CH_4$–$H_2$-air mixtures on the concentration of hydrogen. (**a**) $T_0$ = 293 K (line—calculation by the mechanism [31,32], symbols—experimental values from literary sources: △—[38], o—[40], ◆—[41], ●—[42], ▼—[43], ★—[44], ■—[45]. (**b**) Calculation at $T_0$ (K): *1*—300, *2*—400, *3*—500, *4*—600.

The good agreement of the experimental data and calculation results for the initial temperature $T_0$ = 293 K prompted us to perform similar simulations at higher initial temperatures of the mixture up to $T_0$ = 600 K (Figure 6b). These results allow us to conclude that even at elevated temperatures, with a hydrogen concentration in a methane–hydrogen mixture of less than 40%, it has a weak effect on the burning velocity of methane–air mixtures. A more detailed description of these results is given in [52].

## 6. Discussion

### 6.1. Influence of Temperature and Hydrogen Concentration

A sharp change in the dependence of the activation energy of the autoignition delay time for hydrogen and mixtures with its high concentration at $T \approx 900$ K (Figures 4 and 5) indicates that this temperature should be considered as the boundary between the low-temperature and high-temperature parts of the considered range, near which significant changes in the mechanism of the process occur. This is confirmed by the character of the dependence of the activation energy of the autoignition delay time of stoichiometric methane–hydrogen mixtures on the hydrogen content in them, which is sharply different for this temperature (Figure 7). The curve for $T_0$ = 900 K clearly separates two different process modes with different dependence on the hydrogen concentration. At low temperatures ($T_0$ < 900 K), the activation energy of the autoignition delay time increases monotonically with the hydrogen concentration in the mixture, whereas at high temperatures ($T_0$ > 900 K), it passes through a flat maximum (Figure 7). These temperature regions are separated by a sharply different dependence for $T_0$ = 900 K, in which the activation energy of the autoignition delay time of methane–hydrogen mixtures monotonically increases with the

hydrogen concentration in the mixture from 30.5 kcal/mol for methane to 117.7 kcal/mol for hydrogen, which is almost fourfold (Figure 7).

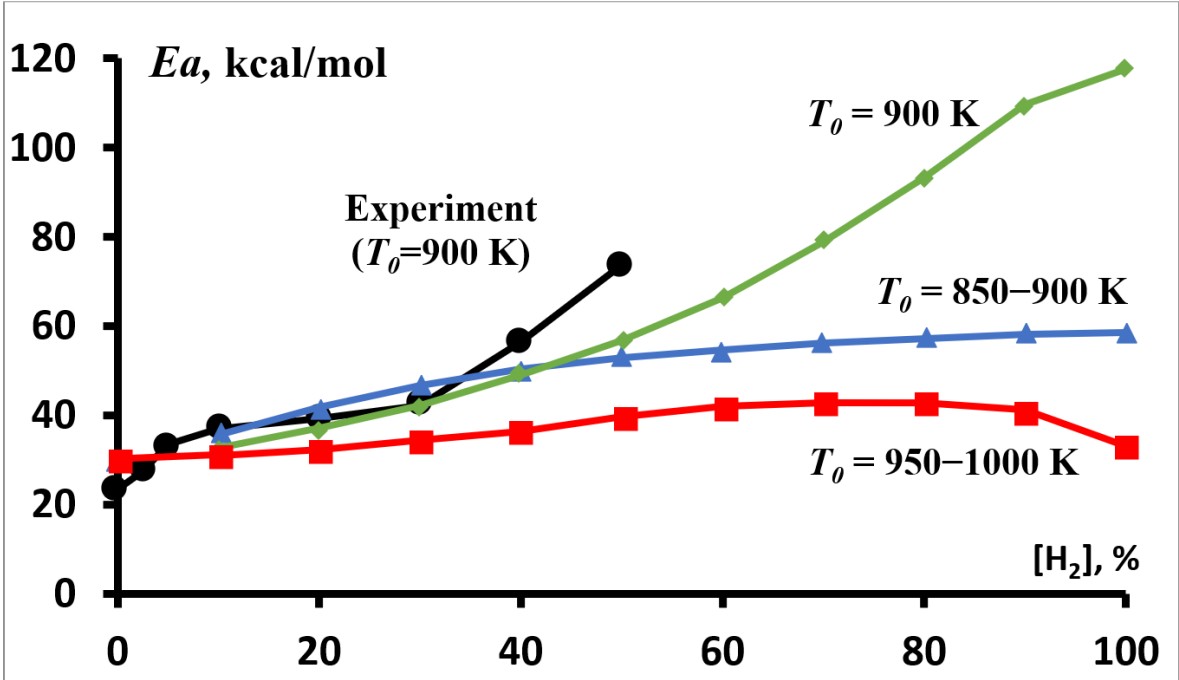

**Figure 7.** Dependence of the effective activation energy $Ea$ for the autoignition delay time of stoichiometric methane–hydrogen–air mixtures on the hydrogen concentration at $P_0 = 1$ atm. •—experimental results ($T_0 = 900$ K). Calculations at $T_0$ (K) 850–900 (▲), 900 (♦) and 950–1000 (■).

The experimental results we obtained for $T_0 = 900$ K (Figure 7), taking into account various factors that can introduce errors [28,29], are in good agreement with the simulation results. When hydrogen concentration in the mixture is changed from zero to 50%, the experimentally determined activation energy of the autoignition delay time of methane–hydrogen mixtures increased monotonically from 23.4 kcal/mol to 73.7 kcal/mol.

The very similar transition in activation energy for hydrogen was also observed in shock tube experiments [24]. At $P = 5$ atm activation energy for hydrogen autoignition time $Ea$ was 39.3 kcal/mol, while at lower temperatures it was equal to 126.9 kcal/mol (Figure 8). At a higher pressure (10 atm), the change in this activation energy for hydrogen was even greater, but with an opposite sign relative to the temperature: from $Ea = 74.5$ kcal/mol at low (~1025 K) temperature to 258 kcal/mol at 1108 K with a subsequent decrease to 49.2 kcal/mol at higher temperatures. A similar behavior was observed at 20 atm (Figure 8).

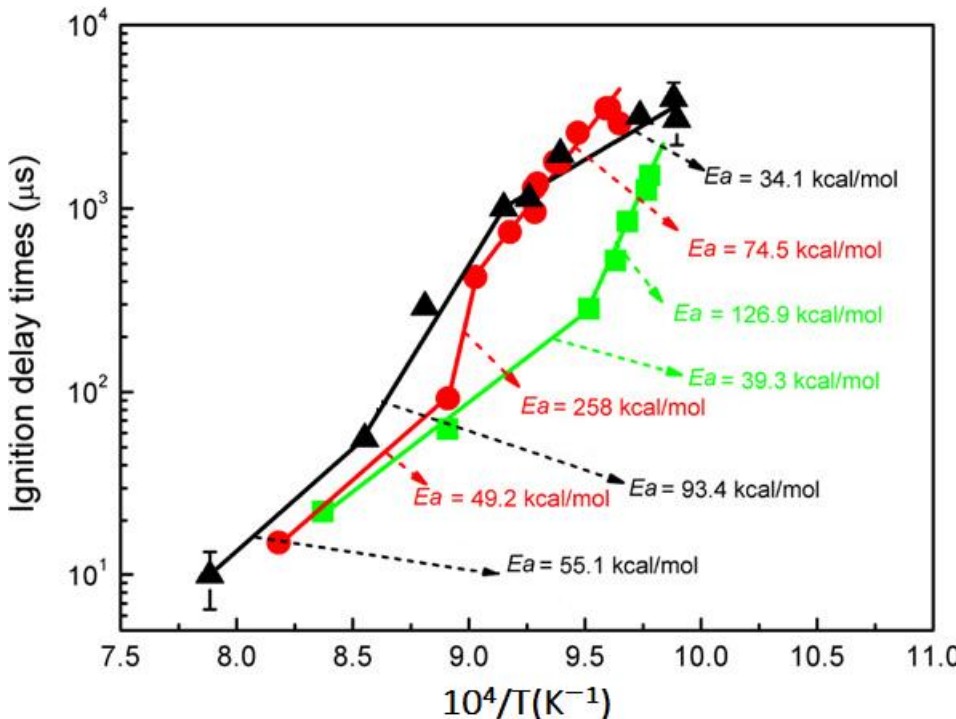

**Figure 8.** Temperature dependence of autoignition delay time for hydrogen at various pressures. (■)—$P$ = 0.5 MPa; (●)—$P$ = 1.0 MPa; (▲)—$P$ = 2.0 MPa (adapted from [24]).

It is worth noting that a significant increase in the activation energy during the transition from the low-temperature region $T < 1000$ K to higher temperatures $T > 1100$ K was also observed in [53] for methane. However, it is possible that the very low values of the autoignition delay time obtained in this work at $T < 1000$ K, which contradict the simulation results, are caused by the inaccuracy of determining the temperature in the shock wave.

*6.2. Effect of Pressure*

The effect of pressure on the autoignition of methane–hydrogen mixtures is complex due to its opposite effect on the autoignition of hydrogen and methane. According to numerous literature data obtained mainly in shock tube experiments for temperatures above 1000 K, an increase in pressure reduces the autoignition delay time of hydrocarbons, including methane. On the contrary, the ignition delays of hydrogen increase with the increase in pressure at temperatures between 1093 and 1170 K. The results from [24] show that at $T$ = 1093 K, the ignition delay of hydrogen at a pressure of 2.0 MPa is ten times longer than that at 0.5 MPa (Figure 8). The complex pressure dependence was also observed in [21].

The capabilities of the experimental equipment we used are limited to an initial pressure of $P_0$ = 3 atm. Taking into account the good agreement of experimental and calculated results for pressures 1 and 3 atm, we found it possible to expand the pressure range to 15 atm during modeling (Figure 9).

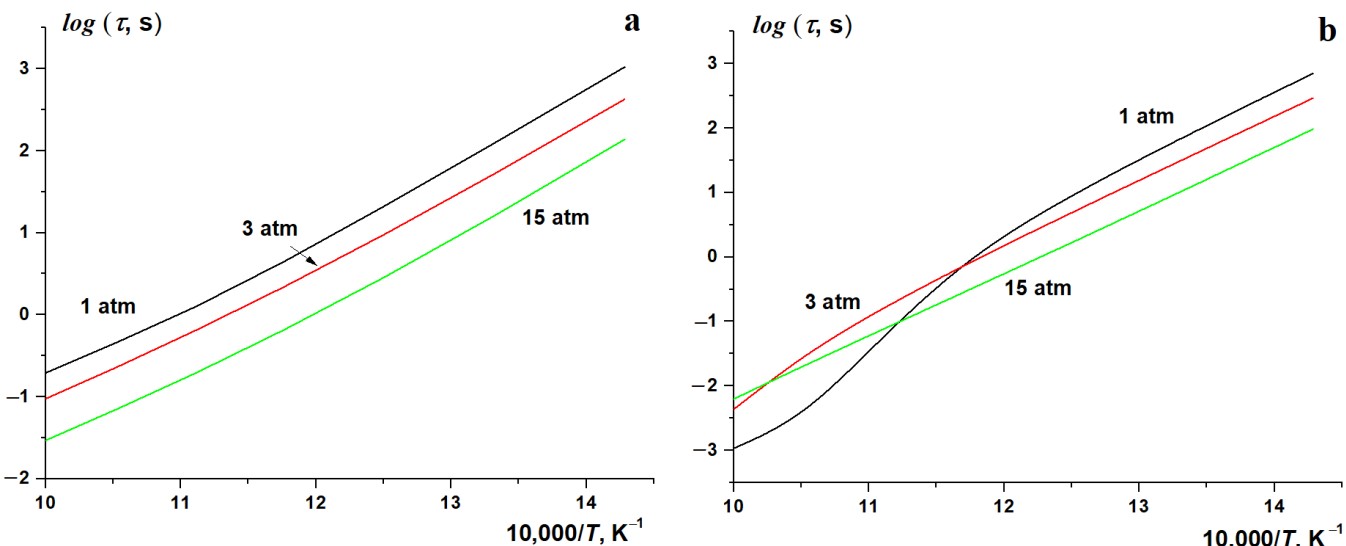

**Figure 9.** Temperature dependence of autoignition delay time for stoichiometric methane–hydrogen–air mixtures at various initial pressures and hydrogen concentrations; (**a**) [$H_2$] = 20%, (**b**) [$H_2$] = 80%.

The results show that for [$H_2$] = 20% in the entire temperature range covered at all pressures used, the temperature dependence of the autoignition delay time of stoichiometric methane–hydrogen mixtures has an Arrhenius form. In the entire temperature range, an increase in pressure promotes the ignition (Figure 9a). This confirms the conclusion that the oxidation of methane–hydrogen mixtures with low hydrogen content occurs mainly by the methane mechanism. However, the results are drastically different for mixtures with [$H_2$] = 80%, the oxidation of which proceeds via the hydrogen mechanism [24,25]. While at $P_0$ = 15 atm, the Arrhenius character of the dependence holds, at $P_0$ = 3 atm, it becomes somewhat distorted, being obviously violated at $P_0$ = 1 atm (Figure 9b). At the same time, while in the low-temperature part of this range, an increase in pressure promotes the autoignition, i.e., reduces the ignition delay time, in its high-temperature part, on the contrary, it inhibits the process. A fundamental change in the nature of the influence of pressure on the ignition process occurs near $T \approx 900$ K. These results are in fairly good agreement with the results presented in Figure 8 [24], showing that, for hydrogen with an increase in pressure, the dependence of the autoignition delay time becomes closer to the Arrhenius expression.

A very similar dependence of the autoignition delay time for hydrogen and methane–hydrogen mixtures with high hydrogen content was observed in shock-tube experiments [21]. In addition to using a different technique and higher temperatures, the authors of [21] performed experiments with methane containing ~8% ethane, which can significantly affect the time and nature of the dependence of the ignition delay on various parameters [28–30], as well as strong dilution (1:5) of the mixture with argon. Nevertheless, in these experiments, the same apparent maximum in the temperature of the autoignition delay time of hydrogen and mixtures with its high content was observed. At a pressure of 1 atm, the temperature of this maximum (~950 K) (Figure 10) is practically identical to the calculated one in Figure 9. As the pressure increases, this maximum shifts to higher temperatures: ~1050 K at *P* = 4 atm and ~1250 K at *P* = 16 atm (Figure 10).

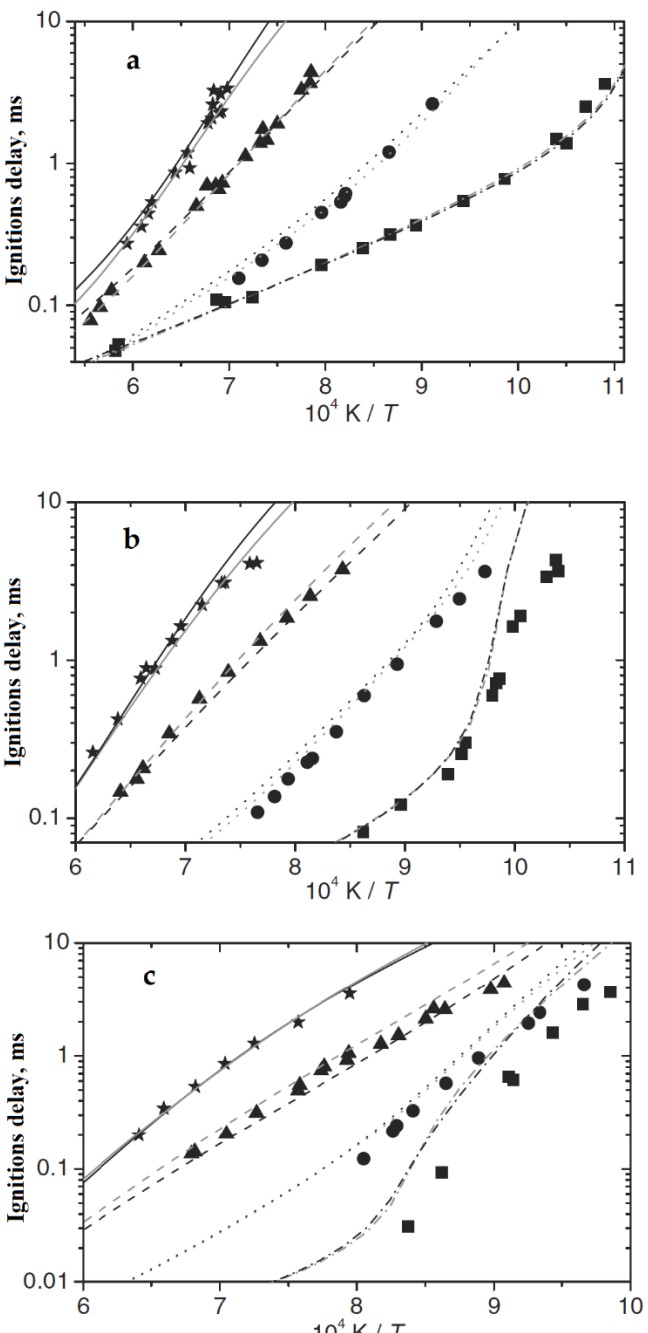

**Figure 10.** Measured and calculated ignition delay times for $H_2$–$CH_4$–$O_2$–Ar mixtures ($\varphi$ = 1.0, 1:5 dilution) at pressures of 1 (**a**), 4 (**b**) and 16 atm (**c**). Experiments: (■)—100% $H_2$, (●)—80% $H_2$, (▲)—40%, (★)–0% $H_2$. Lines represent the results of kinetic simulations (adapted from [21]).

A different effect of pressure on the autoignition delay time of methane–hydrogen–air mixtures at different temperatures is clearly seen in Figure 11, which demonstrates the calculated dependence of the autoignition delay time of stoichiometric $CH_4$–$H_2$–air mixtures at $P_0$ = 15 atm on the initial temperature at various concentrations of hydrogen. While at high temperatures, an increase in the concentration of hydrogen promotes autoignition, at low temperatures it clearly, albeit slightly, inhibits it. This is quite consistent with the experimental results presented in Figure 2, which show that, at low temperatures $T \approx 850$ K, hydrogen very weakly promotes the autoignition of methane; however, the promoting effect of hydrogen increases with the temperature. If desired, one can even see the presence

of a small inhibitory effect of hydrogen in Figure 2, but the spread of experimental results is too large for such an unambiguous conclusion.

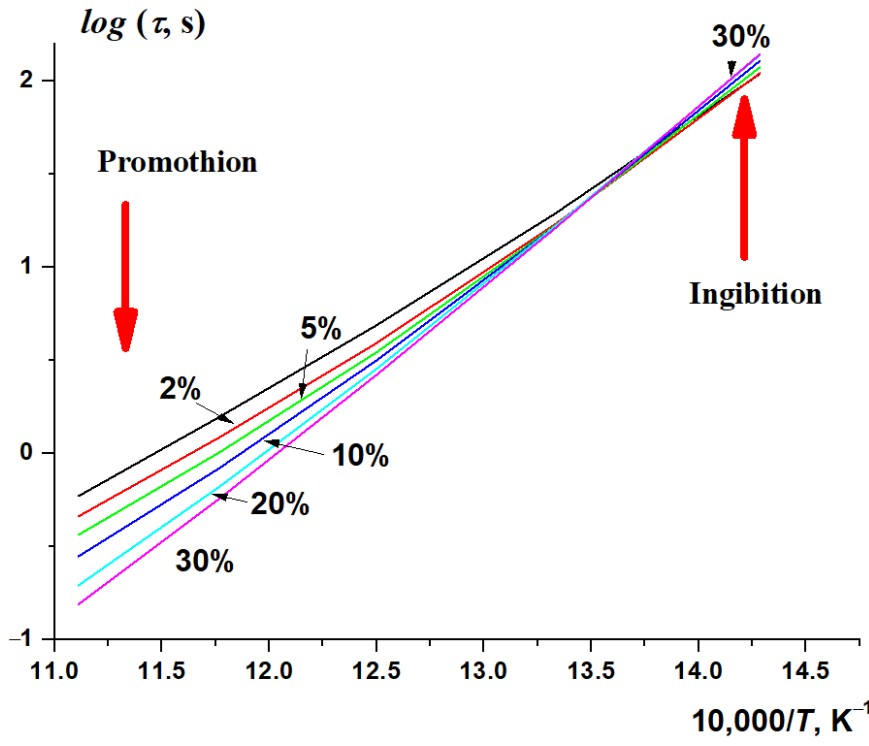

**Figure 11.** Calculated dependence of the autoignition delay time of stoichiometric $CH_4$–$H_2$–air mixtures on the initial temperature at different hydrogen concentrations (%); $P_0$ = 15 atm.

The calculated dependence of the autoignition delay time of methane–hydrogen–air mixtures with different hydrogen concentration on the pressure in the range from 1 to 15 atm at $T_0$ = 900 K is shown in Figure 12. With low hydrogen concentration, the autoignition delay time monotonically decreases with increasing pressure. However, for hydrogen itself and for mixtures with its high concentration, pressure increases the autoignition delay time of methane–hydrogen–air mixtures at low pressures, leading to the appearance of a maximum at $P \approx 3$ atm. Only at higher pressures is a monotonous decrease in the autoignition delay time is observed.

Similar effects were observed in [24] for hydrogen. At slightly higher temperatures, above 1000 K, an increase in hydrogen pressure from 5 to 20 atm led to a noticeable increase in the ignition delay time and a change in the ignition delay activation energy (Figure 8). At even higher temperatures, above 1100 K, apparently due to the final changeover to the high-temperature mechanism of hydrogen oxidation, both its autoignition delay time and the activation energy of its change no longer depend on pressure (Figure 8).

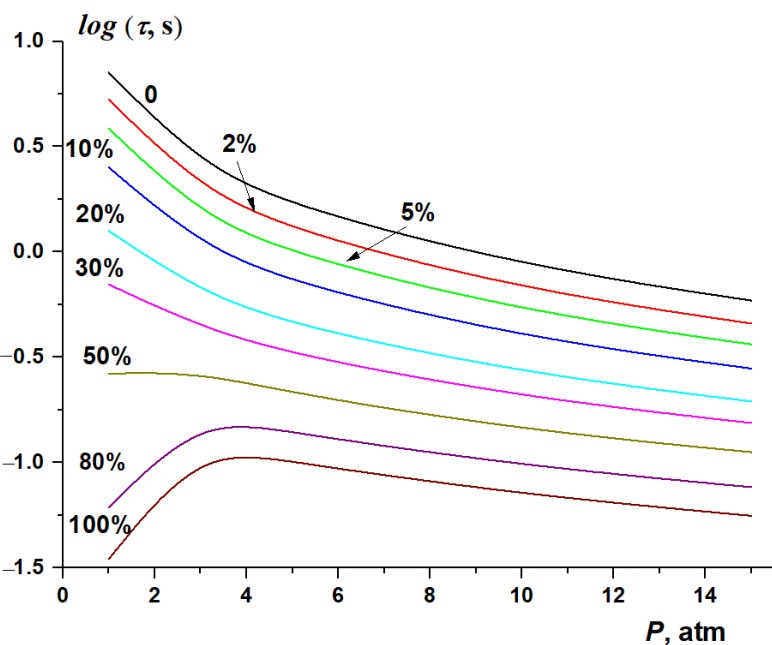

**Figure 12.** Dependence of the autoignition delay time of stoichiometric of $CH_4$–$H_2$–air mixtures on the initial pressure at $T_0 = 900$ K and various hydrogen concentrations.

*6.3. Kinetic Interpretation of Observed Phenomena*

A kinetic interpretation of the results obtained is presented below. The fact that for a stoichiometric methane–air mixture, the activation energy of the autoignition delay time in the studied temperature range is practically constant and significantly lower than the activation energy (~57 kcal/mol) of the chain initiation reaction,

$$CH_4 + O_2 \rightarrow CH_3^\bullet + HO_2^\bullet, \tag{1}$$

can be explained by the branched-chain nature of the process in this temperature range. It is worth noting that for the branched-chain process of the partial oxidation of very rich mixtures of methane at similar temperatures, an activation energy of 46 kcal/mol was experimentally obtained [27,54], whereas for rich mixtures it was in the range of 42.8–48.4 kcal/mol [24]; that is, also significantly lower than the activation energy of reaction (1).

At temperatures below 900 K, the activation energy of the ignition delay time increases with the hydrogen concentration in the mixture (Figure 7). This can formally be interpreted as a manifestation of the inhibitory effect of hydrogen on the methane autoignition in this temperature range. A similar phenomenon of inhibition by hydrogen was observed for the autoignition of rich methane–propane mixtures [55]. Significant distinctions in the behavior of mixtures with high and low hydrogen content is probably due to substantial differences in the low-temperature ($T < 900$ K) mechanisms of methane and hydrogen oxidation.

At temperatures below 900 K, methyl peroxy radicals $CH_3OO^\bullet$, formed in the equilibrium reaction

$$CH_3^\bullet + O_2 \leftrightarrow CH_3OO^\bullet \tag{2}$$

play a leading role in the oxidation of methane. They lead to the formation of methyl hydroperoxide $CH_3OOH$ and the subsequent degenerate chain branching when it decays into radicals

$$CH_3OOH \rightarrow CH_3O^\bullet + OH^\bullet, \tag{3}$$

due to which, at temperatures below 900 K, methane oxidation proceeds as a fast branched-chain process [27] with an effective activation energy noticeably lower than the energy of bond rupture in reaction (1). However, at temperatures above 900 K, the equilibrium in

reaction (2) shifts to the left, the rate of formation of methyl peroxy radicals and, accordingly, methyl peroxide drops sharply, and the process ceases to be chain-branched, which leads to a decrease in its rate, with the reaction passing into the region of negative temperature coefficient (NTC) of the reaction rate. At even higher temperatures, the branched-chain process of methane oxidation is realized by a different, high-temperature mechanism [15,27]. Therefore, the temperature range near 900 K is transitional from low-temperature to high-temperature methane oxidation.

Coincidentally, a similar temperature range is also transitional for the mechanism of hydrogen oxidation, although the reasons here are different. Radical generation reaction during hydrogen oxidation

$$H_2 + O_2 \rightarrow H^\bullet + HO_2{}^\bullet, \tag{4}$$

is similar to the reaction (1) and yields $HO_2{}^\bullet$ and $H^\bullet$ radicals, while the latter reacts with $O_2$ to produce $HO_2{}^\bullet$, similar to reaction (2). However, unlike methyl peroxy radicals, hydroperoxy radicals $HO_2{}^\bullet$ are inactive at temperatures below 900 K, being consumed mainly by recombination

$$HO_2{}^\bullet + HO_2{}^\bullet \rightarrow H_2O_2 + O_2 \tag{5}$$

Reaction (5) is a chain-termination step, since hydrogen peroxide, unlike methyl peroxide, is relatively stable at these temperatures, because its decay through the reaction

$$H_2O_2 \rightarrow OH^\bullet + OH^\bullet, \tag{6}$$

similar to reaction (3), does not provide fast enough chain-branching at these temperatures. Note that the rate of the other chain-branching reaction in the hydrogen oxidation mechanism,

$$H^\bullet + O_2 \rightarrow OH^\bullet + O^{\bullet\bullet}, \tag{7}$$

is also too low at these temperatures. Therefore, the oxidation of hydrogen at low temperatures proceeds as an unbranched-chain process, and its addition to the system may even lead to an inhibition effect due to the additional consumption of methyl radicals by the reactions

$$CH_3{}^\bullet + H_2 \rightarrow CH_4 + H^\bullet , \tag{8}$$

$$H^\bullet + O_2 + M \rightarrow HO_2{}^\bullet + M, \tag{9}$$

leading eventually to hydrogen peroxide (reaction (5)), a relatively stable species under these temperatures. Apparently, this can explain the effect of ignition of oxygen-rich methane–propane mixtures by hydrogen additives [55], and a threefold increase in the effective activation energy observed by us as the concentration of hydrogen in methane–hydrogen–air mixtures was increased (Figure 7). Figure 13 presents the main routes of reactants' conversion in the chain mechanism of ignition of $CH_4$–$H_2$–air mixtures at various temperatures and hydrogen concentrations.

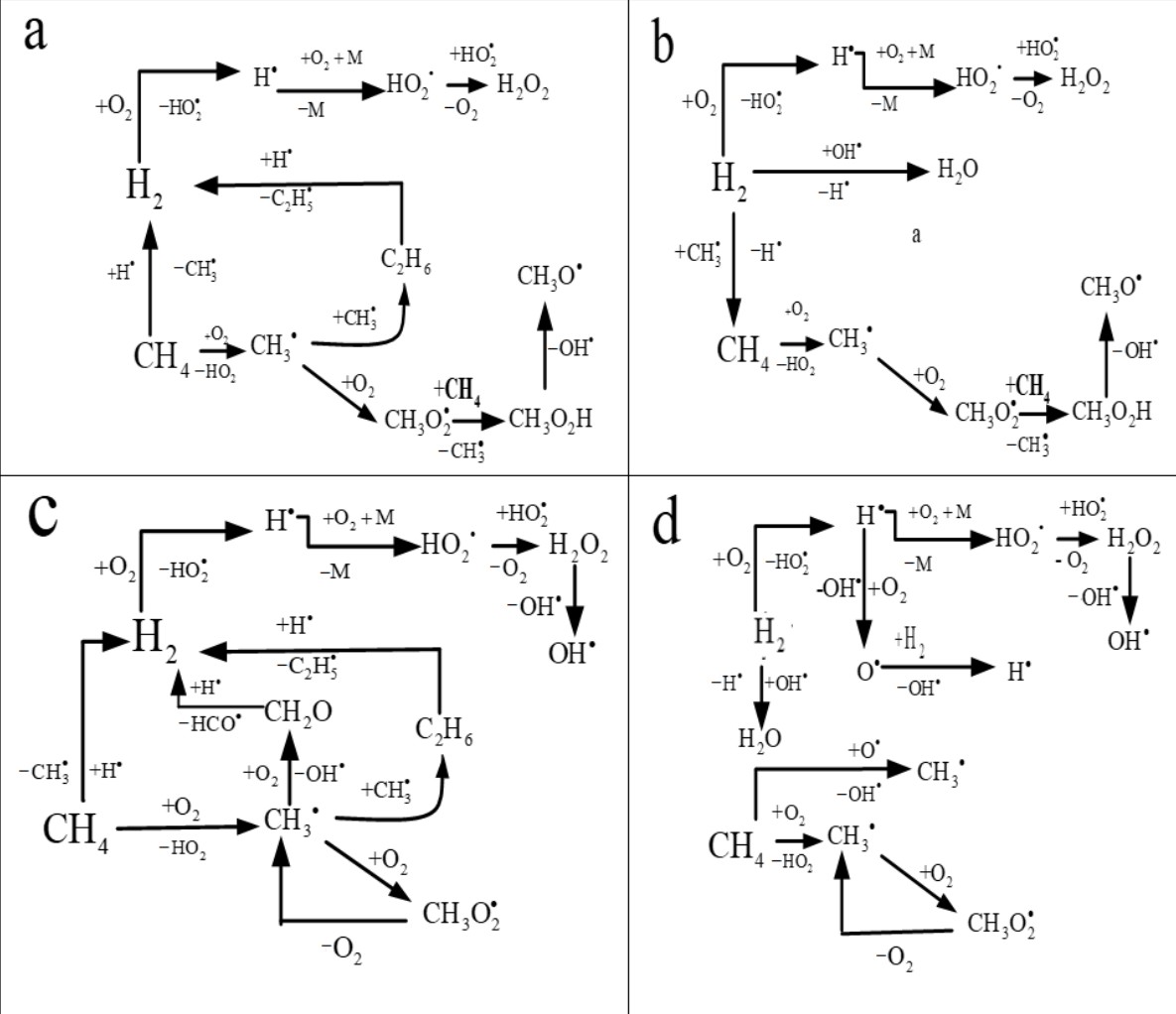

**Figure 13.** Main routes of conversion of reagents in the chain mechanism of ignition of mixtures $CH_4$-$H_2$-air. (**a**)—[$H_2$] = 20%, $T_0 < 900$ K; (**b**) [$H_2$] = 80%, $T_0 < 900$ K; (**c**)—[$H_2$] = 20%, $T_0 > 900$ K; (**d**)—[$H_2$] = 80%, $T_0 > 900$ K.

At $T_0 < 900$ K, when present in low concentrations, hydrogen promotes the removal of active radicals produced during the branched-chain oxidation of methane due to hydrogen peroxide formation (Figure 13a). At high hydrogen concentrations, on the contrary, hydrogen oxidation is promoted by branched-chain oxidation of methane (Figure 13b). At $T_0 > 900$ K and a low concentration of hydrogen, conjugate radical oxidation processes of methane and hydrogen occur, in which hydrogen promotes the oxidation of methane (Figure 13c). At $T_0 > 900$ K and a high concentration of hydrogen, despite the total pool of radicals in the system, their oxidation proceeds practically independently (Figure 13d). An indirect confirmation of this interpretation can be a rapid increase in the maximum concentration of hydrogen peroxide with an increase in the initial hydrogen concentration during the oxidation of methane-hydrogen mixtures at low (about 800 K) temperatures (Figure 14).

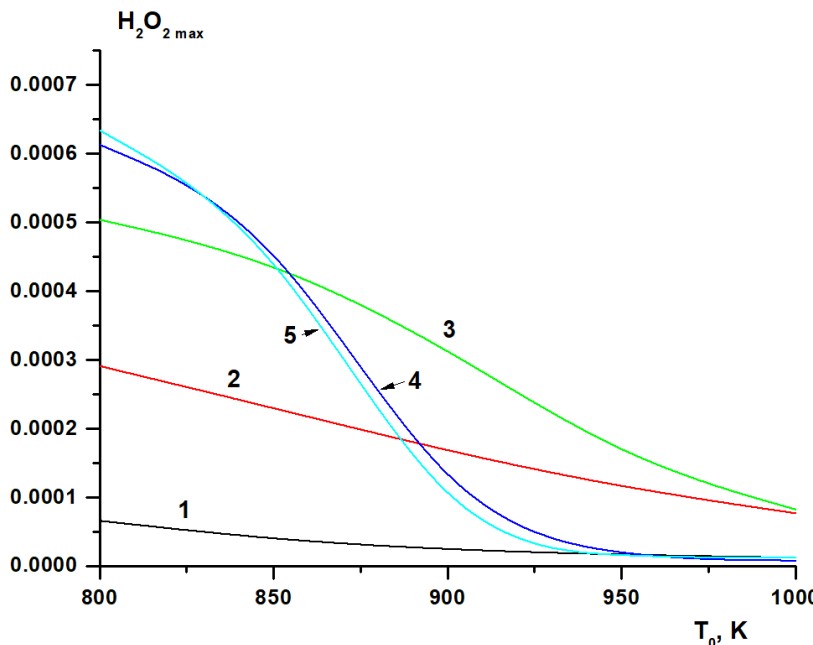

**Figure 14.** Calculated dependence of the maximum concentration of hydrogen peroxide (molar fractions) on the initial temperature at various hydrogen concentrations (%): 0 (*1*), 40 (*2*), 70 (*3*), 90 (*4*), 100 (*5*).

As the temperature increases, the activity of hydroperoxide radicals $HO_2^{\bullet}$ and the rate of decomposition of hydrogen peroxide increase. In the vicinity of $T \approx 900$ K, the rate of decomposition of hydrogen peroxide becomes quite high, and its maximum concentration rapidly decreases with increasing temperature (Figure 14). Therefore, at temperatures above 900 K, the recombination of $HO_2^{\bullet}$ radicals ceases to inhibit the process, which leads to a significant change in the mechanism of the oxidation of hydrogen and mixtures with its high content. This temperature region of the fundamental change in the mechanism is known in the literature as "$H_2O_2$ turnover" [56]. In addition, with increasing initial temperature, the role of branching reaction (7) grows rapidly so that hydrogen oxidation becomes a branched-chain process. With high hydrogen content, this manifests itself as a rapid decrease in the effective activation energy of the autoignition delay time. The changes in the mechanism of hydrogen oxidation described above lead to the appearance of a maximum in the temperature dependence of the autoignition delay time of hydrogen and mixtures with its high content (Figure 5).

The unique property of methane to provide a branched-chain oxidation at relatively low temperatures fundamentally distinguishes its low-temperature oxidation not only from the oxidation of hydrogen, but also from the oxidation of its closest homologues. Therefore, despite the higher bond energy and, accordingly, the higher activation energy of radical generation, the effective activation energy of the ignition delay time of methane oxidation is lower than that of hydrogen, ethane, propane, or even butane [28–30]. However, at temperatures above 900 K, the rate of formation of methyl peroxy radicals and their role in the oxidation of methane turn out to be insignificant, and therefore, the main features of the mechanisms of oxidation of methane and hydrogen become much similar, which is reflected similar values of the activation energy for the autoignition delay time of methane, hydrogen, and mixtures thereof at $T > 1000$ K (Figure 5). It is quite natural that due to a somewhat lower H–H bond energy compared to the $CH_3$–H bond energy, the activation energy for the ignition delay time of hydrogen and mixtures with its high content in this region is lower than that of methane.

As the methane-to-hydrogen ratio in the mixture changes, so does their role in the oxidation mechanism. At a low initial concentration of hydrogen in the mixture, as a result of a rapid branched-chain process, after the ignition the methane concentration

quickly drops to zero. Note that the concentration of hydrogen, which in this case is an intermediate product of oxidation, increases, reaching a maximum, which is almost a third of the initial methane content. Then, due to the end of the branched-chain process, the hydrogen concentration rapidly decreases to a certain steady-state value as a result of achieving thermodynamically equilibrium composition of the products (Figure 15, curve 1).

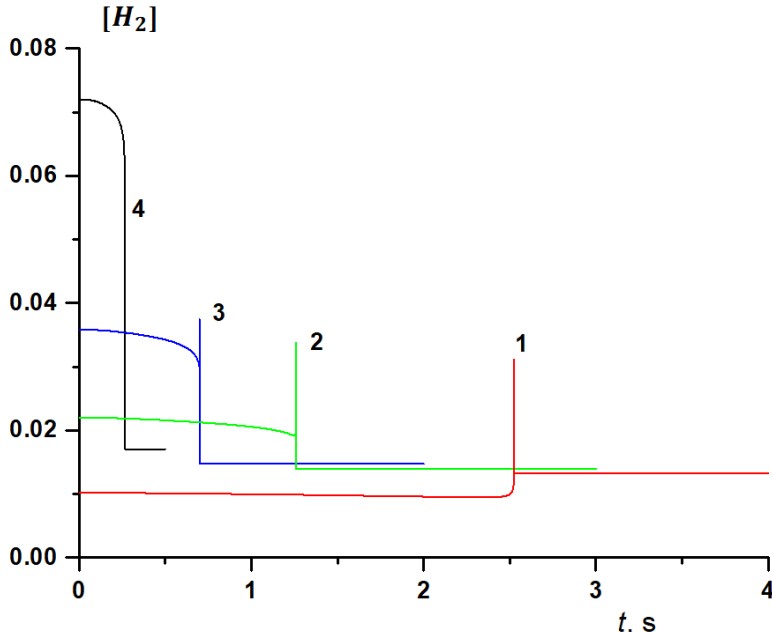

**Figure 15.** The kinetics of hydrogen concentration (mol. fractions) at $T_0$ = 900 K and the initial concentration of hydrogen in the methane-hydrogen mixture $[H_2]_0$ (%): 10 (*1*), 20 (*2*), 30 (*3*), 50 (*4*).

However, while the kinetics of changes in the methane concentration remains qualitatively the same over the entire range of the initial temperatures $800 \leq T$ (K) $\leq 1000$ and hydrogen concentration in the fuel from 0 to 90%, the behavior of hydrogen changes significantly as its content in the mixture changes. Figure 15 shows the calculated change in the kinetics of the hydrogen concentration at $T_0$ = 900 K and various values $[H_2]$. At $10 \leq [H_2]$ (%) $\leq 30$, after a slight gradual decline before ignition, a very sharp spike in the hydrogen concentration is observed at the moment of ignition, which then rapidly decreases to a certain stationary value. With an increase in the fraction of $H_2$ in the initial mixture, the peak hydrogen concentration decreases, and its stationary final concentration increases slightly. At $[H_2]$ = 10%, the decrease in the fraction of hydrogen before ignition is almost imperceptible; however, with increasing $[H_2]$, the difference between its initial concentration and that reached by the time of ignition increases. At $[H_2]$ = 40% (not shown in Figure 15), the peak of the hydrogen concentration is barely noticeable, and at $[H_2]$ = 50% (Figure 15, curve 4), the peak is not visible at all, while the change in hydrogen concentration becomes similar to the change in methane concentration, remaining so with a further increase in the initial hydrogen content. With a change in temperature, the picture does not change qualitatively, but the peak hydrogen concentration increases with an increase in the initial temperature.

The peak concentration of hydrogen is explained by the competition of the processes of its formation and consumption, and with its absence or low initial concentrations in the fuel at the initial stage of the process its formation clearly prevails. As the initial fraction of hydrogen in the mixture increases, while the fraction of methane, respectively, decreases, the rate of hydrogen oxidation begins to prevail from the very beginning over its formation in secondary reactions, and the peak on its kinetic curve disappears.

The ignition delay times of the H$_2$ subsystem are determined by the two competing reactions:

$$H + O_2 (+ M) \leftrightarrow HO_2 (+ M) \tag{10}$$

and

$$H + O_2 \leftrightarrow OH + O. \tag{11}$$

Reaction (10) becomes dominant at higher pressures, whereas reaction (11) becomes dominant at higher temperatures because of its high activation energy. If reaction (10) is considerably faster than reaction (11), the ignition delay times are increased because less chain branching occurs by reaction (11). Therefore, the ignition of the pure hydrogen system at 16 atm and $T < 1100$ K is slower than at 4 and 1 atm because reaction (10), which is close to the low-pressure limit, is about 4 or 16 times faster at the higher pressure [21]. This exceeds the effect of the higher absolute concentrations due to the higher pressure which dominates at higher temperatures and in hydrocarbon systems.

## 7. Conclusions

The complex influence of hydrogen concentration, temperature, and pressure on the autoignition of methane–hydrogen mixtures requires the detailed analysis of these factors. The results of the present study suggest that simplified approaches to the interpretation of the autoignition of such mixtures, based on additivity rules for the properties of their components, are unacceptable. In addition to the safety issues of working with methane–hydrogen mixtures, the obtained results call into question the expediency of the practical use of methane–hydrogen mixtures as a reference scale for determining the detonation characteristics of hydrocarbon gas-engine fuels. Currently, for this purpose, the calculation of their methane number (MN) is widely used on a scale in which the detonation resistance of methane is taken as 100, and hydrogen as 0 [57–59]. However, the behavior of methane–hydrogen mixtures in internal combustion engines will strongly depend on the mode of its operation, and not all modes provide behavior similar to that of methane–alkane mixtures. The results of this work show that just in the range of conditions corresponding to the ignition of the mixture in ICEs [26], the assumption of the identity of the autoignition of methane–alkane and methane–hydrogen mixtures is not fulfilled, on the use of which the methane scale is based. Apparently, methane numbers, if they can be used at all, can be applicable exclusively for comparing the detonation resistance of various mixtures of methane with its heavier homologues. At high enough concentrations of hydrogen and compounds of other classes in the mixture, one can hardly expect adequate results by applying the methane scale to them.

On the other hand, the present work confirms the conclusion drawn in [24,25,47,52,60] that at a hydrogen concentration of less than 40%, its presence has a rather weak effect on the propensity to ignite and the combustion characteristics of methane–hydrogen mixtures (Figures 3 and 6). This allows us to consider this concentration as an upper limit of hydrogen content in a mixture with methane, which provides the possibility of using existing gas equipment and established safety rules for the transportation and practical use of methane–hydrogen mixtures.

**Author Contributions:** V.A.: conceptualization, writing, supervision; A.B.: kinetic simulations; A.A.: kinetic simulations; K.T.: experimental investigations; A.N.: experimental investigations. All authors have read and agreed to the published version of the manuscript.

**Funding:** This work was performed within the framework of the Program of Fundamental Research of the Russian Federation. Reg. № 122040500068-0.

**Data Availability Statement:** Not applicable.

**Conflicts of Interest:** The authors declare no conflict of interest.

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
