# Peer review of "Autoignition of Methane–Hydrogen Mixtures below 1000 K"

_processes, doi:10.3390/pr10112177_

Round 1

Reviewer 1 Report

This is a very well written paper in my opinion.

The scope is clearly stated and the content is well organized. Conclusions appear to be sound, consistent and clear.

Only a couple of minor comments from my side. They are concerned with the experimental procedure for the ID measurement. 

1. Row 136: the authors mention a settling time of 48 hours for homogeneization. It seems quite a lot to me for a volume of 10x10 cm. Can the authors justify this? What about the risk of blend stratification?

2. A timem history of the pressure would be useful for the reader to support the understanding of the measurement procedure.

Also, a direct comparison between the pressure and the H2 concentrations, such as those in Fig. 15, would be a significant link between the experimental and the numerical activities.

Author Response

Response to Reviewer 1

Comment 1:

Row 136: the authors mention a settling time of 48 hours for homogeneization. It seems quite a lot to me for a volume of 10x10 cm. Can the authors justify this? What about the risk of blend stratification?

Response:

Here, apparently through our fault, there is a slight misunderstanding. We are talking about holding the mixture not in a reactor, but in a steel cylinder with a volume of 10 liters, in which it is prepared at high pressure. It takes at least a day to fully mix the mixture under these conditions. The text has been clarified.

Comment 2:

A time history of the pressure would be useful for the reader to support the understanding of the measurement procedure.

Response:

We agree with the Reviewer in the usefulness of such information. Typical pressure change curves were presented by us in already published works [28, 29] containing a more detailed description of the experimental technique. Since there are a lot of figures in this paper, we consider it inappropriate to demonstrate once more the time history of the pressure here, but added the references to these previous works.

Comment 3:

Also, a direct comparison between the pressure and the H2 concentrations, such as those in Fig. 15, would be a significant link between the experimental and the numerical activities.

Response:

We agree that such information would be very useful, but unfortunately our experimental equipment does not allow us to register the kinetics of changes in the hydrogen concentration during ignition, which can only be traced by calculation.

Authors are grateful to the Reviewer for the careful analysis of our work.

Reviewer 2 Report

This is the interesting paper on the topical problem.

I found just one drawback, however the significant one.

Authors perform the kinetic modelling based on the detailed mechanism NUL. First, the analyze the temperature dependence of the time delay which is the primary kinetic characteristic of the complex reaction. Some depencences are non-trivial, i.e. non-additive.  Then, they explain the observed dependences in terms of steps of the detailed mechanism. So, such interpretation is done within the body of paper.

My strong recommendation is: such interpretation must be presented very clear both in the abstract and in conclusion as well. The maximum of "Time delay-Temperature" must be explained kinetically as a result of the interplay  of some elementary steps. 

The role of step HO2 +HO2 should be explained in more detail. I have some concern about the significance of this step because it is the reaction between two radicals of small concentration. This statement must be supported by HO2 conconcentration.

Authors use the term "Arrhenius dependence" discussing the dependence of the time delay on the temperature. It is not rigorous. The Arrhenius dependence is about the dependence of the kinetic coefficient on the temperature (the temperature is "up", and "kinetic coefficicient is"up") Qualitatively, the time delay is a reciprocal kinetic coefficient ((the temperature is "up", and "time deay" is "down")

Better to use the term "temperature dependence of the time delay" 

There is some mistake in Fig 11 ( see words "prohibition" , "inhibition")

Author Response

Response to Reviewer 2

Comment 1:

I found just one drawback, however the significant one.

Authors perform the kinetic modelling based on the detailed mechanism NUL. First, the analyze the temperature dependence of the time delay which is the primary kinetic characteristic of the complex reaction. Some depencences are non-trivial, i.e. non-additive. Then, they explain the observed dependences in terms of steps of the detailed mechanism. So, such interpretation is done within the body of paper.

My strong recommendation is: such interpretation must be presented very clear both in the abstract and in conclusion as well. The maximum of "Time delay-Temperature" must be explained kinetically as a result of the interplay of some elementary steps.

Response:

Perhaps we did not understand the meaning of this remark accurately enough, but the entire theoretical part of the work is based on the use of detailed kinetic models, the calculation of which has long been a mandatory and routine practice for describing the complex radical kinetics of ignition and combustion processes. Such a kinetic description is practically mandatory, and allows not only to interpret experiments on ignition of various gas systems, but also to predict their behavior in other conditions with high accuracy. Therefore, it would be strange to specially emphasize the use of kinetic modeling methods in the modern study of ignition processes. And "Time delay – Temperature" relationship cannot be expressed as a sequence of “some successive elementary steps”, since it is a consequence of the complex interaction of several hundred elementary steps running in parallel, the interplay between which changes with changes in external conditions and constantly changes in the course of the process. A simplified graphical interpretation of the most important routes for the most typical cases, obtained as a result of kinetic modeling, is presented in Fig. 13.

Comment 2:

The role of step HO2 +HO2 should be explained in more detail. I have some concern about the significance of this step because it is the reaction between two radicals of small concentration. This statement must be supported by HO2 concentration.

Response:

The significant role of HO2 radicals in the processes of low-temperature oxidation, combustion and ignition has been established for a very long time. References to some relevant works [15, 27, 57] are given in the text. A more detailed description of their role would require a significant increase in the volume of manuscript. Their main role is that at these low temperatures they react too slowly, accumulate in high concentrations and decay mainly by the recombination reaction (5), that is, their formation actually leads to the termination of radical-chain process. This is a long and well-established fact, see, e.g., references mentioned above. But with an increase in temperature, the rate of other reactions with their participation increases, and this leads to a change in the mechanism of the process. This process is described in detail on page 15 and is confirmed by the results of kinetic modeling, the graphical interpretation of which is presented in Fig. 13.

Comment 3:

Authors use the term "Arrhenius dependence" discussing the dependence of the time delay on the temperature. It is not rigorous. The Arrhenius dependence is about the dependence of the kinetic coefficient on the temperature (the temperature is "up", and "kinetic coefficicient is"up") Qualitatively, the time delay is a reciprocal kinetic coefficient ((the temperature is "up", and "time deay" is "down")

Better to use the term "temperature dependence of the time delay"

Response:

According to the terminology accepted in chemical kinetics, the Arrhenius dependence, which should not be identified with the Arrhenius law, is any linear dependence in Arrhenius coordinates that is “log W” and “1/T”, independent of the sign of its slope. For example, in well-known examples of the existence of a region of a negative temperature coefficient (NTC) of the reaction rate, as well as in a number of elementary trimolecular reactions, the reaction rate decreases with increasing temperature. These dependences are still described by the Arrhenius expression, but with negative activation energy. In all publications on the investigation of ignition delay, a linear decrease in the ignition delay time with temperature in Arrhenius coordinates is considered to be Arrhenius dependence, which is a direct consequence of the increase in the rate of the reaction with temperature in accordance with the Arrhenius law.

Comment 4:

There is some mistake in Fig 11 (see words "prohibition", "inhibition")

Response:

We did not find any error in Figure 11. There is no term "prohibition" there. According to the results of kinetic modeling, at low temperatures, hydrogen inhibits methane ignition, therefore for this region the term "inhibition" is indicated, and at high temperatures, on the contrary, it promotes methane ignition, therefore for this region the term "promotion" is indicated.

Authors are grateful to the Reviewer for the careful analysis of our work and useful discussion.

Round 2

Reviewer 2 Report

No comments